# Can alcohol consumption in Germany be reduced by alcohol screening, brief intervention and referral to treatment in primary health care? Results of a simulation study

Jakob Manthey[1,2,3]*, Adriana Solovei[4], Peter Anderson[5,6], Sinclair Carr[2], Jürgen Rehm[2,7,8,9,10]

1 Institute of Clinical Psychology and Psychotherapy, Technische Universität Dresden, Dresden, Germany, 2 Department of Psychiatry and Psychotherapy, Center for Interdisciplinary Addiction Research (ZIS), University Medical Center Hamburg-Eppendorf (UKE), Hamburg, Germany, 3 Medical Faculty, Department of Psychiatry, University of Leipzig, Leipzig, Germany, 4 Department of Health Promotion, CAPHRI Care and Public Health Research Institute, Maastricht University, Maastricht, The Netherlands, 5 Population Health Sciences Institute, Newcastle University, Newcastle upon Tyne, United Kingdom, 6 CAPHRI Care and Public Health Research Institute, Maastricht University, Maastricht, The Netherlands, 7 Institute for Mental Health Policy Research & Campbell Family Mental Health Research Institute, Centre for Addiction and Mental Health, Toronto, Ontario, Canada, 8 Dalla Lana School of Public Health, University of Toronto, Toronto, ON, Canada, 9 Department of Psychiatry, University of Toronto, Toronto, Ontario, Canada, 10 Department of International Health Projects, Institute for Leadership and Health Management, I.M. Sechenov First Moscow State Medical University, Moscow, Russian Federation

* jakob.manthey@tu-dresden.de

**Data Availability Statement:** All relevant data are within the manuscript and its Supporting Information files.

## Abstract

### Background

Screening, brief intervention and referral to treatment (SBIRT) is a programme to reduce alcohol consumption for drinkers with high alcohol consumption levels. Only 2.9% of patients in primary health care (PHC) are screened for their alcohol use in Germany, despite high levels of alcohol consumption and attributable harm. We developed an open-access simulation model to estimate the impact of higher SBIRT delivery rates in German PHC settings on population-level alcohol consumption.

### Methods and findings

A hypothetical population of drinkers and non-drinkers was simulated by sex, age, and educational status for the year 2009 based on survey and sales data. Risky drinking persons receiving BI or RT were sampled from this population based on screening coverage and other parameters. Running the simulation model for a ten-year period, drinking levels and heavy episodic drinking (HED) status were changed based on effect sizes from meta-analyses.

In the baseline scenario of 2.9% screening coverage, 2.4% of the adult German population received a subsequent intervention between 2009 and 2018. If every second PHC

**Funding:** The research leading to these results or outcomes has received funding from a) the European Horizon 2020 Programme for research, technological development and demonstration (Grant Agreement no. 778048) – "Scale-up of Prevention and Management of Alcohol Use Disorders and Comorbid Depression in Latin America" (SCALA), b) the National Institute on Alcohol Abuse and Alcoholism (Grant Agreement no. 1R01AA024443-01A1) – "Calibrated agent simulations for combined analysis of drinking etiologies (CASCADE)", c) the National Institute on Alcohol Abuse and Alcoholism (Grant Agreement no. 1R01AA028224-01) – "Evaluation of the impact of alcohol control policies on morbidity and mortality in Lithuania and other Baltic states" (ALC-LTU), and d) the German Federal Ministry of Health (grant number ZMVl1-2517DMS227) – "Implementierung und Evaluation der S3-Leitlinie zu Screening, Diagnose und Behandlung alkoholbezogener Störungen" (IMPELA). URLs of funding bodies: https://www.niaaa.nih.gov/ https://ec.europa.eu/programmes/horizon2020/ The funding was received by: a) SCALA: JM, AS, PA, SC, JR b) CASCADE: JR c) ALC-LTU: JM & JR d) IMPELA: JM, SC & JR The views expressed here reflect those of the authors only and the funders are not liable for any use that may be made of the information contained therein. The funders had no role in study design, data collection and analysis, decision to publish, or preparation of the manuscript.

**Competing interests:** The authors have declared that no competing interests exist.

patient would have been screened for alcohol use, 21% of adult residents in Germany would have received BI or RT by the end of the ten-year simulation period. In this scenario, population-level alcohol consumption would be 11% lower than it was in 2018, without any impact on HED prevalence. Screening coverage rates below 10% were not found to have a measurable effect on drinking levels.

## Conclusions

Large-scale implementation of SBIRT in PHC settings can yield substantial reductions of alcohol consumption in Germany. As high screening coverage rates may only be achievable in the long run, other effective alcohol policies are required to achieve short-term reduction of alcohol use and attributable harm in Germany. There is large potential to apply this open-access simulation model to other settings and for other alcohol interventions.

## Introduction

Globally, alcohol use is a major risk factor for the burden of mortality and disease [1,2]. Various initiatives have been set by the World Health Organization (WHO) and the United Nations has set objectives to reduce this burden [3]. Among five high-impact and cost-effective strategies to reduce alcohol use and the resulting burden of disease, the WHO recommends facilitating access to screening and brief interventions (SBI), and treatment [4–6].

Screening for alcohol use, brief intervention (if the alcohol use patterns exceed a certain threshold; BI), and referral to specialized treatment (RT; full acronym: SBIRT) is an evidence-based practice used to identify, reduce, and prevent risky alcohol and other drug use and attributable harm (for an introduction, see [7,8]; for overviews, see [9,10]).

In Germany, application of SBIRT is recommended by the 'Guidelines on Screening, Diagnosis and Treatment of Alcohol Use Disorders' [11], however, survey data from the federal state of Bremen suggest that only 2.9% of patients were screened by their primary health care (PHC) providers in 2016 [12]. Given the persistently high prevalence of alcohol use disorders (2018: men: 9.2%; women: 3.6%, [13]), alcohol per capita consumption (APC; recorded sales in 2018: 10.8 litres, [14]), and alcohol-attributable mortality (2016: 45,000 or 5% of all deaths, [15]) in Germany, strategies to curb consumption and adverse consequences are urgently required and SBIRT presents a viable option.

In a 2014 systematic review, 22 studies have been identified estimating the cost-effectiveness of SBI programmes, none of which was performed for Germany [16]. The only application of a simulation model to quantify the effects of SBI in Germany known to the authors was carried out as part of a report issued by the Organization for Economic Co-operation and Development [17]. While the report found the potential of SBI to be sizeable, the results cannot be easily reproduced or the method extended, as the underlying programme was not published.

In fact, none of the 22 studies identified in the 2014 review disclosed their simulation programmes [16]. Further, the two perhaps most common and internationally applied simulation programmes do not disclose their source code to the public (for applications of the programs, see [18] and [19]), making it impossible for other researchers to apply or adapt the programme on their own.

In this contribution, we develop an open-access simulation model which serves to estimate the impact on increased SBIRT activities in PHC settings on population drinking patterns and

levels. We first give an extensive description of the simulation model and then present results of an application for Germany, testing how alcohol consumption would have changed if more patients would have been screened for their alcohol use in PHC settings.

## Materials and methods

The methods employed can be summarized in four steps: 1) A complete time series of alcohol data, more specifically of drinking status and drinking levels, stratified by sex, age, and educational status, was obtained by combining survey and sales data for the years 2009 to 2018; 2) A hypothetical population for the year 2009 was drawn, for which drinking status, as well as drinking levels, were assigned based on data from the first step. 3) For risky drinking persons (definition see Table 1) who were identified by their treating general practitioners through alcohol screening in that year, effects of BI and RT were applied to change drinking levels and patterns. 4) We continued with step 3 in the next year but prior to applying effects from BI and RT, we accounted for: a) attenuating effects of BI and RT over time; and, b) secular changes in prevalence of any drinking and heavy episodic drinking (HED). The simulation procedure is outlined in Fig 1 and all simulation parameters are summarized in Table 1.

### Step 1: Obtaining a complete time series of alcohol data

A complete time series of alcohol data was obtained by combining survey, alcohol sales, as well as population data. From a repeated large-scale cross-sectional survey conducted in 2009, 2012, 2015, and 2018 (key survey results reported in [13,21]), we obtained prevalence estimates for each drinking status (lifetime abstinence, former drinking—FD, and current, i.e., past-

**Table 1. Description and sources of simulation parameters.**

| Type of parameter | Description of parameter | Source |
|---|---|---|
| APC | Recorded APC for all years 2009 to 2018 | obtained from [20] |
| Drinking status | Prevalence of drinking status at base year, stratified by sex, age group, and educational level | obtained from [13,21] |
| Risky drinking | Cutoffs to define low, medium and high risk drinking based gram pure alcohol intake per day (women: 21g and 41g; men: 41g and 61g) | obtained from [22] |
| PHC access | Probability for PHC visits, stratified by sex, age group, educational level, and drinking level | calculated from [23] |
| Screening probability | Probability of screening among PHC patients: baseline (2.9%) but varied in alternative scenarios (0%, 10%, 25%, 50%, 75%) | baseline obtained from [12] |
| Intervention probability | Probability of BI and RT among positively screened patients: 50% | obtained from [12] |
| Effect size of brief intervention | The proportional reduction of a) daily drinking levels (women: -15.8%, -31.1 to -1.1%; men: -12.0%, -18.6 to -5.7%) and b) risk reduction of HED (risk difference: -0.07, -0.12 to -0.02) | a) recalculated and b) obtained from a meta-analysis [24]. |
| Effect size of referral to treatment | Effect size of RT, assumed to be the same as for BI | |
| Attenuation of effects | The attenuation of intervention effects, assuming that BI effects to remain stable for a period of four years and to attenuate thereafter and reach 0 after ten years (linearly imputed for all years five to ten) | according to [25] and [26] |

APC = pure alcohol per capita consumption; BI = Brief Intervention; HED = Heavy Episodic Drinking;

PHC = Primary Health Care; RT = Referral to Treatment.

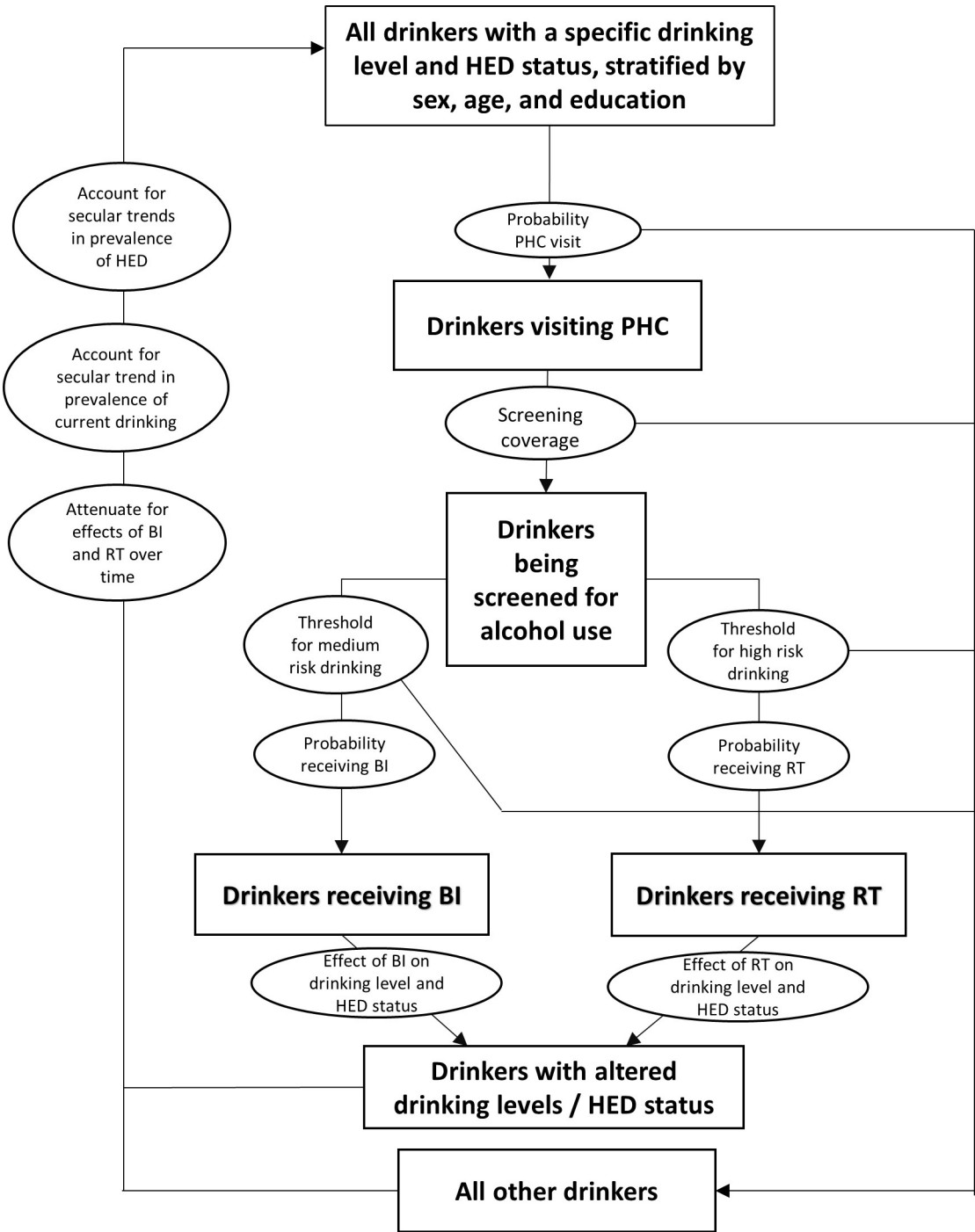

**Fig 1. Flow-chart of simulation procedure.** Round shapes represent simulation parameters; rectangular boxes represent the sample selected based on specified parameters; the loop repeats every year, for a period of 10 years.

year, drinking—CD), heavy episodic drinking (HED, defined as at least one occasion on which at least 60g pure alcohol was consumed in the past 30 days), as well as reported daily intake of pure ethanol (in grams per day, g/d). All data were stratified by sex (women, men), age group (15–34, 35–49, 50–64, 65+ years), and educational level (low, middle, and high

according to the International Standard Classification of Education [27]). HED prevalence was further stratified by drinking level, for low, medium, and high-risk drinkers. As risk thresholds, we referred to the WHO definition of low chronic risk (women: below 21 g/d, men: below 41 g/d), medium chronic risk (women: between 21 and 41 g/d, men: between 41 and 61 g/d), and high chronic risk (women: at least 41 g/d, men: at least 61 g/d [28]). Missing years in survey data were linearly imputed. Because surveys are prone to under-reporting real drinking levels (for an overview of under-reporting drinking levels in European surveys including the one used, see [29]), we referred to recorded APC from the World Health Organization for the years 2009 to 2018 [20], indicating the pure alcohol sold per adult per year in Germany. The APC was disaggregated for each sex-age-education group accounting for differences in survey-reported prevalence of current drinking and reported drinking levels, resulting in the average drinking levels per drinker in each group of interest. Lastly, sex, age, and education stratified population data for all years was obtained by combining population data from UN (providing data for all age groups and years, but not by education) and EUROSTAT (providing education breakup for all years, but only for age groups up to 74 years). As a result, we obtained a complete set of prevalence estimates for lifetime abstinence, former drinking, current drinking, and HED, as well as average drinking levels per drinker–for all years 2009 to 2018 and by sex, age, and education. These data served as input data for the next step.

## Step 2: Simulating baseline alcohol consumption in the population

For the year 2009, a population sample of 100 persons was drawn, stratified by sex, age, and educational level. Using binomial distributions, the drinking status prevalence estimates described in step 1 were used to determine the drinking status (either lifetime abstainer, former drinker, or current drinker) for each person. Second, we determined daily drinking levels (in grams pure alcohol per day) for each current drinker, which was drawn from a gamma distribution, which has been shown to approximate alcohol use self-reports from surveys [30,31]. Lastly, HED status was determined for each current drinker based on the data from step 1, again using binomial distributions.

## Step 3: Applying effects of SBIRT

In order to apply the effects of SBIRT, the following four conditions had to be fulfilled: Persons had to: attend PHC; be screened for alcohol use; drink riskily; and receive a BI (for medium risk drinkers) or RT (for high-risk drinkers). In the following, each consecutive conditional step is described in detail.

First, the prevalence of at least one annual PHC visit was obtained from a large-scale survey conducted in 2016 [23], assuming that these prevalences did not change over time. As with the alcohol data, the PHC visit prevalence estimates were also stratified by sex, age, education, and risky drinking status (low, medium, high; thresholds see above). Based on these prevalence estimates, binomial distributions were used to determine whether a person had at least one PHC visit in the current year.

Second, the likelihood to be screened was set at 2.9% in the baseline scenario (as reported in a recent survey, [12]). In alternative scenarios, we repeated the simulation at screening rates of 0%, 10%, 25%, 50%, and 75%. This parameter served as the main manipulation for the simulation–as all other parameters were kept constant. For drinking persons attending PHC, binomial distributions with the screening rate were used to determine whether a person was screened for their level of alcohol use or not.

Third, for positively screened persons, i.e., being screened and drinking above the medium or high risk thresholds, we assumed that every second person (regardless of sex, age,

education) would effectively receive a BI or RT, respectively (based on a recent survey of general practitioners [12]).

Fourth, for positively screened persons receiving an effective intervention, the drinking levels were reduced based on effect sizes determined in a 2018 meta-analysis [24]. Rather than using the reported absolute reduction of drinking levels, we applied the proportional reduction of drinking levels from baseline to account for the above-mentioned under-reporting (see Table 1 for effect sizes). In addition to manipulating the drinking levels for positively screened persons receiving BI or RT, the HED status was also changed based on effect sizes from the same meta-analysis (see Table 1 for effect sizes).

### Step 4: Accounting for secular changes and attenuating intervention effects

In this ten-year model, step 3, i.e., the application of SBIRT effects, was repeated for each year. However, prior to applying the effects in each year, we accounted for a) attenuating intervention effects from previous years, and b) secular changes in APC and drinking status prevalence. First, we assumed that intervention effects on drinking levels remained stable for a period of four years (according to [25]), attenuate thereafter, and nullify after ten years (according to [26], linearly imputed for all years five to ten). For drinkers giving up HED following BI or RT, we assumed that the chance to re-engage in HED was 50% chance starting from the second year post intervention. Second, we corrected drinking levels among drinkers and drinking status to match the observed trajectories in the input data.

### Sensitivity analysis

Keeping all other parameters constant, we tested the impact of the attenuation of intervention effects over time. In an additional sensitivity analysis, we assumed that any intervention effect diminishes three years post intervention.

### Reporting the simulation findings

We simulated six scenarios: one baseline, or as-is scenario, with an annual screening rate of 2.9%, and five alternative scenarios, with annual screening rates of 0%, 10%, 25%, 50%, and 75%. The outcomes of interest were drinking levels and prevalence of HED, which are the two alcohol exposure variables known to be most impacted by BI and RT delivered in PHC settings.

All findings are reported against the baseline scenario for the final year 2018, thus describing the cumulative effects over a ten-year period. The simulations and all other analyses were performed in R version 4.0.5 [32]. All confidence intervals were estimated by repeating the simulation for 100 times, i.e., re-running the ten-year simulation for 100 different population samples and obtaining the 95% percentiles to estimate the degree of uncertainty around point estimates. The variation was the result of drawing parameters from their respective confidence intervals (CI; e.g. current drinking prevalence for low-educated women aged 15 to 34 in 2009 ranged between 76.2% and 86.1%), except for recorded APC (no variation assumed for sales statistics). The complete R code including input data is appended to this paper to allow for complete reproducibility and adjustment of parameters to other settings (see S1 File).

This simulation study did not involve any human subjects, thus, no ethical review was sought and no participant consent was obtained. As input data for the simulation, we obtained aggregated and fully anomized secondary data from previous surveys, which have undergone formal ethical reviews (for details, see [21,23]).

## Results

### Alcohol exposure in Germany in the baseline scenario

Between 2009 and 2018, alcohol consumption hardly changed in Germany. In this period, APC remained largely constant at around 11 litres pure alcohol, and the prevalence of current drinking slightly decreased, which was more pronounced among lower educated persons (from 77.9% to 73.9%) than among higher educated persons (from 92.9% to 90.2%). In 2018, 73.8% of drinkers were estimated to have low risk drinking levels, while medium and high-risk drinking was present among 11.9% and 14.2% of drinkers. Prevalence of HED in 2018 among all adults was estimated at 34.9%. For 2009, all alcohol consumption estimates by sex, age, and education are presented in S1 Table and for all other years available in S1 File.

### The simulation explained

Fig 2 illustrates the simulation using the drinking level trajectories of three individual drinkers, sampled from the group of women aged 35 to 49 years with medium educational level. The green line was taken from the scenario without any screening activity, thus, the drinking level follows the observed trend of drinking levels in this population, as specified in the input data. The blue line was taken from the baseline scenario, thus, at a screening rate of 2.9%. This individual had a very similar trajectory of their drinking levels as the first individual–however, in 2012, they were screened and received an intervention (as indicated by the vertical line), thus, reducing their drinking level. For the remaining years, drinking levels were again parallel to the green line, reflecting the secular trend in this population. Lastly, the brown line was taken from the scenario of 75% screening activity, and this individual received three interventions in the ten-year simulation period.

### Estimated coverage of SBIRT

In the baseline, i.e., the as-is scenario, an estimated 21.0% (95% CI: 19.3% to 22.7%) of the adult population was screened at least once for alcohol use within the ten-year period, based

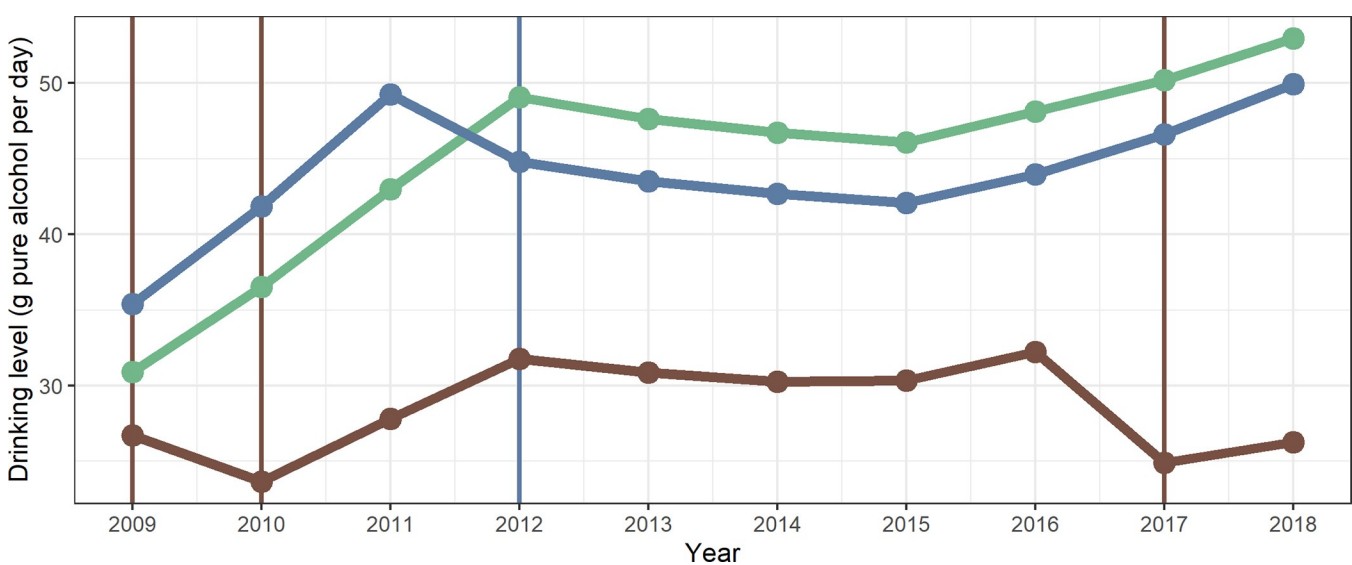

**Fig 2. Trajectory of daily drinking levels for three individuals.** Each colour represents one individual and the vertical lines indicate reception of an intervention in that year (see text for explanation).

on an annual screening rate of 2.9%. If general practitioners screened every 10[th] person, about every second adult (57.1%, 95% CI: 54.8% to 59.0%) would have their alcohol use measured by the end of the decade (at 50% screening rate: 99.1%, 95% CI: 98.7% to 99.4%).

Across the ten-year period, every 40[th] adult (2.5%, 95% CI: 1.8% to 3.1%) was estimated to have benefitted from an intervention, i.e., BI for medium risk and RT for high-risk drinkers, following alcohol screening in PHC in the baseline scenario. Increasing the screening rates to 10%, 50%, or 75% would have increased the intervention rates to 7.6% (95% CI: 6.7% to 8.8%), 21.1% (95% CI: 20.0% to 22.3%), or 23.7% (95% CI: 22.2% to 25.2%), respectively.

The screening and intervention rates achieved by end of the ten-year period are illustrated in Fig 3.

## Impact on APC and HED

In Tables 2 and 3, the simulation results are presented for two outcomes of interest, the mean daily drinking levels and the prevalence of HED in the adult population. Comparing the 2018 estimates of each alternative to the baseline scenario, the results suggest that mean daily drinking levels would not significantly differ if alcohol screening rates varied between 0% and 10%. If alcohol use was assessed in every fourth patient, reductions in drinking levels among men and the youngest age groups could be achieved. At 50% and 75% screening rates, reductions in drinking levels in the entire population could have been achieved, which would be driven mainly by men, younger and older drinkers (see Table 2).

In Figs 4 and 5, four trajectories of APC are presented by age and educational level, for the baseline and three scenarios with higher annual alcohol screening coverage. As illustrated in Fig 4 for women, the APC trajectories and their corresponding confidence intervals show a high overlap in most groups, suggesting no significant impact of SBIRT on drinking levels.

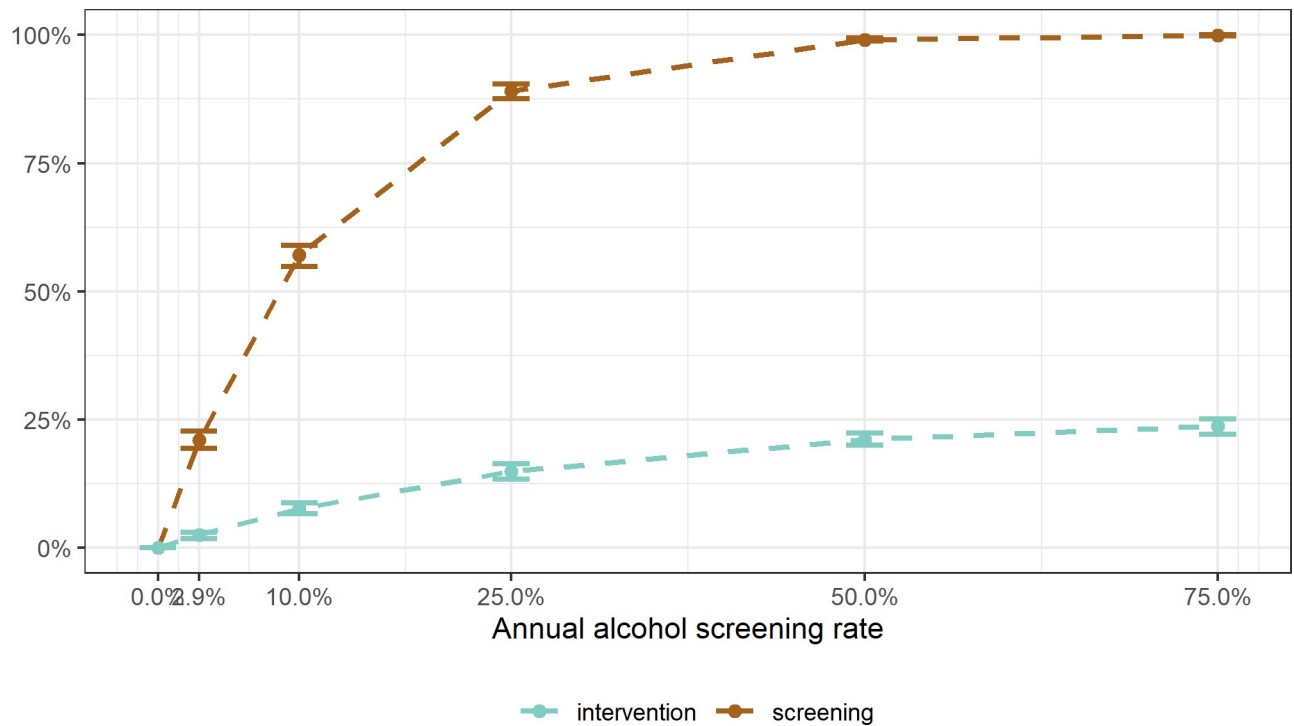

**Fig 3. Alcohol screening and intervention rates at the end of the ten-year simulation period, depending on annual alcohol screening rate in German primary health care settings.** The brown line is the percentage of the German population who was screened for their alcohol use. The green line is the percentage who received either a brief intervention (for medium-risk drinking) or were referred to treatment (for high-risk drinking).

**Table 2. Differences in mean drinking levels in different alcohol screening rate scenarios, as compared to baseline scenario at the end of the ten-year simulation period.**

| Screening coverage | 0% | 10% | 25% | 50% | 75% |
|---|---|---|---|---|---|
| Total population | 1.0% (-3.5 to 10.8%) | -1.2% (-5.0 to 6.6%) | -5.6% (-8.9 to 2.1%) | **-11.4% (-15.2 to -4.4%)** | **-17.0% (-20.5 to -10.0%)** |
| Women | 0.8% (-6.9 to 20.2%) | -0.9% (-7.4 to 14.6%) | -5.0% (-11.3 to 9.4%) | -10.4% (-16.3 to 2.6%) | **-14.9% (-20.4 to -1.9%)** |
| Men | 1.1% (-1.0 to 2.9%) | -1.6% (-4.1 to 0.9%) | **-6.0% (-8.3 to -3.1%)** | **-13.0% (-16.1 to -9.9%)** | **-19.8% (-22.2 to -17.1%)** |
| 15–34 | 1.2% (-1.9 to 5.6%) | -2.5% (-6.6 to 3.4%) | **-7.3% (-10.7 to -1.3%)** | **-15.7% (-19.4 to -10.8%)** | **-22.6% (-26.5 to -18.0%)** |
| 35–49 | 0.8% (-6.9 to 15.7%) | -0.6% (-9.7 to 29.0%) | -4.3% (-11.1 to 12.6%) | -7.9% (-17.2 to 12.0%) | -12.8% (-22.1 to 0.9%) |
| 50–64 | 0.6% (-11.9 to 40.6%) | -0.3% (-10.6 to 18.3%) | -4.0% (-14.9 to 19.2%) | -8.4% (-20.3 to 12.2%) | **-12.2% (-22.6 to -4.8%)** |
| 65–99 | 1.0% (-7.8 to 15.7%) | -2.4% (-7.9 to 6.8%) | -6.8% (-15.6 to 7.4%) | **-13.7% (-21.5 to -1.2%)** | **-20.3% (-26.9 to -11.3%)** |
| Low | 0.7% (-4.6 to 22.6%) | -1.4% (-6.9 to 23.7%) | -5.3% (-10.5 to 4.1%) | **-11.4% (-16.9 to -2.5%)** | **-16.8% (-21.8 to -8.7%)** |
| Medium | 1.0% (-6.5 to 17.6%) | -2.0% (-8.8 to 11.3%) | -6.6% (-11.2 to 4.3%) | **-11.9% (-19.0 to -1.5%)** | **-18.3% (-24.1 to -5.7%)** |
| High | 0.8% (-3.5 to 6.3%) | -1.6% (-5.1 to 9.8%) | **-4.9% (-8.4 to -0.4%)** | **-10.4% (-13.7 to -5.3%)** | **-14.9% (-19.0 to -12.0%)** |

Note. Presented are relative changes of mean drinking levels to the as-is-scenario, with a screening rate of 2.9%. 95% confidence interval in brackets. Bold results indicate confidence intervals not overlapping with 0, indicating significant differences to the baseline scenario. Low/Medium/High refers to educational status.

However, significant reductions of drinking levels could have been achieved for four out of twelve subgroups if one out of two PHC patients were screened for their alcohol use: 15 to 34 year olds with low and middle education levels, 35 to 49 year olds with high education levels, and 65 to 99 year olds with high education levels.

For men (Fig 5), APC trajectories showed greater deviation across the screening scenarios. If every second PHC patient was screened for their alcohol use, reductions in all except for four out of twelve subgroups could have been achieved: 35 to 49 year olds with low or middle education levels, and 50 to 64 year olds with low or middle education levels.

For HED prevalence, the simulation results suggest that reductions could not have been achieved is most scenarios and population groups. Only at a 75% screening rate, modest decreases could have been achieved among men (see Table 3).

## Sensitivity analyses

In sensitivity analyses, we modeled the impact of SBIRT under the more conservative assumption according to which the intervention effects would completely diminish three years post intervention, as compared to the slower attenuation beginning only five years post intervention

**Table 3. Differences in prevalence of heavy episodic drinking in different alcohol screening rate scenarios, as compared to baseline scenario at the end of the ten-year simulation period.**

| Screening coverage | 0% | 10% | 25% | 50% | 75% |
|---|---|---|---|---|---|
| Total population | 1.4%(-4.5 to 8.2%) | 0.8%(-5.6 to 7.4%) | -0.9%(-7.3 to 6.0%) | -3.0%(-10.9 to 4.6%) | -5.2%(-13.0 to 1.4%) |
| Women | 2.0%(-9.7 to 13.5%) | 1.1%(-10.7 to 13.5%) | -1.1%(-12.6 to 11.1%) | -3.5%(-16.4 to 10.5%) | -3.8%(-20.8 to 7.9%) |
| Men | 0.4%(-3.2 to 5.9%) | 0.4%(-4.1 to 5.0%) | -1.2%(-5.2 to 4.1%) | -3.4%(-8.6 to 1.3%) | **-6.0%(-10.4 to -0.7%)** |
| 15–34 | 1.0%(-6.7 to 9.9%) | -1.0%(-7.0 to 7.0%) | -2.2%(-7.6 to 5.2%) | -4.2%(-10.5 to 4.7%) | -7.9%(-14.4 to 0.7%) |
| 35–49 | 0.6%(-13.0 to 19.2%) | 1.4%(-11.1 to 17.2%) | -0.6%(-14.5 to 15.9%) | -2.6%(-17.1 to 18.0%) | -3.6%(-18.9 to 14.5%) |
| 50–64 | 2.5%(-13.8 to 18.5%) | 0.5%(-12.1 to 24.9%) | -0.4%(-14.3 to 18.1%) | -2.3%(-18.3 to 14.1%) | -3.2%(-17.4 to 10.1%) |
| 65–99 | -0.6%(-9.5 to 21.4%) | 0.2%(-11.2 to 13.8%) | -1.8%(-13.9 to 10.4%) | -5.6%(-16.9 to 7.7%) | -7.8%(-18.2 to 6.4%) |
| Low | 0.5%(-7.2 to 11.5%) | -0.6%(-8.0 to 9.4%) | -1.1%(-10.7 to 7.9%) | -3.6%(-12.1 to 7.8%) | -5.6%(-13.3 to 6.4%) |
| Medium | 2.2%(-7.5 to 11.9%) | 0.5%(-8.9 to 13.3%) | -0.7%(-9.9 to 11.6%) | -3.4%(-14.9 to 7.6%) | -4.9%(-15.9 to 5.1%) |
| High | 0.0%(-7.8 to 13.3%) | -0.1%(-10.0 to 10.7%) | -2.4%(-10.1 to 8.2%) | -3.3%(-14.2 to 9.5%) | -4.7%(-15.9 to 5.8%) |

Note. Presented are relative changes to the as-is-scenario, with a screening rate of 2.9%. 95% confidence interval in brackets. All confidence intervals overlap with 0, thus indicating no significant difference to the baseline scenario. Low/Medium/High refers to educational status.

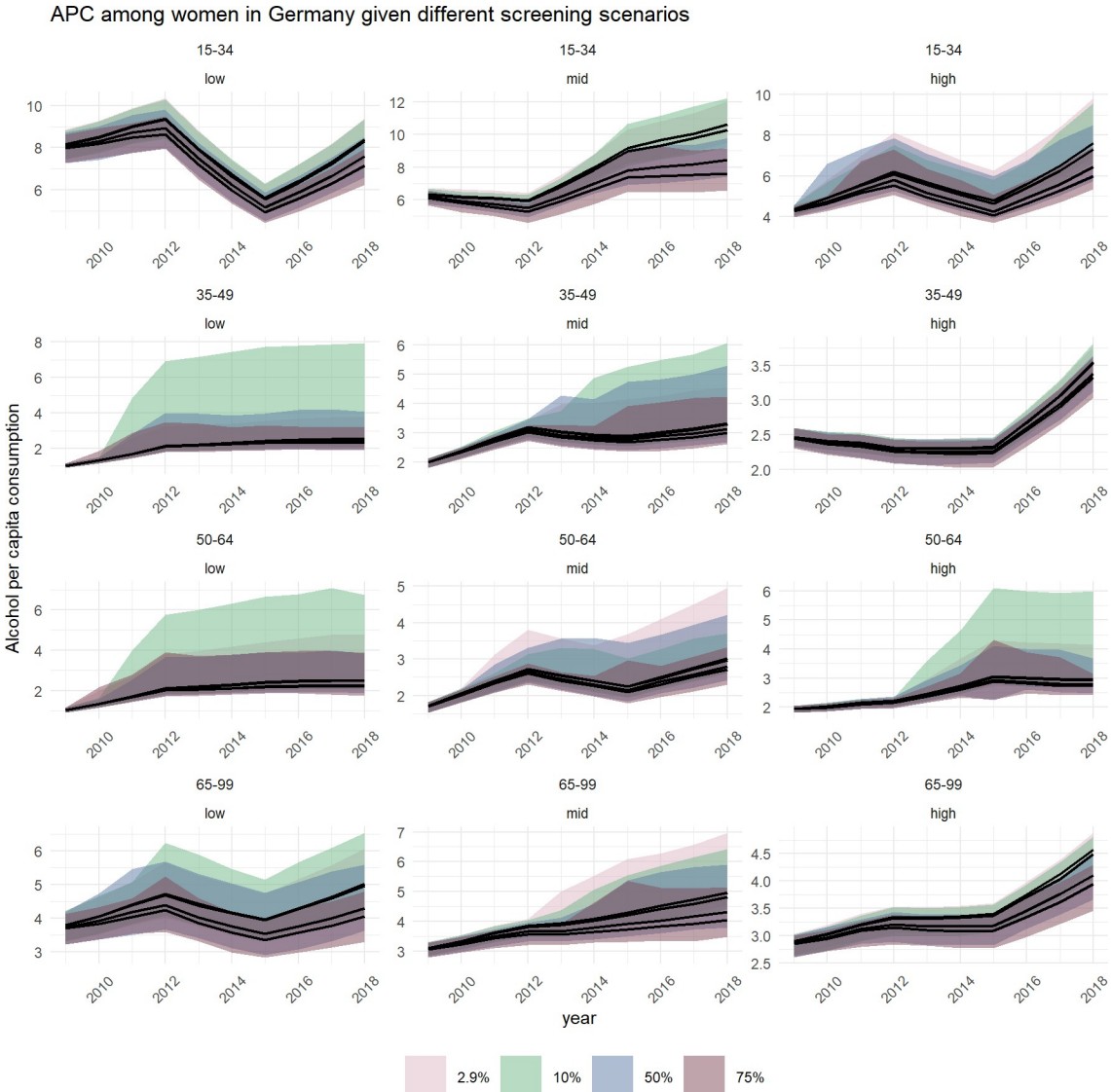

**Fig 4. Trends of alcohol per capita consumption (in litres pure alcohol) among German women between 2009 and 2018, stratified by age (horizontal) and education status (vertical), by annual screening coverage (coloured shade).** In each facet, the upper line represents the baseline scenario with a screening rate of 2.9%. The increasing coverage rate is represented by all lower lines with the second highest representing 10%, the third highest representing 50%, and the lowest representing 75% screening coverage rate.

as implemented in the main analyses. As illustrated in Fig 6, the scenarios of 0 to 25% screening coverage are robust to the underlying assumption, i.e., changing the assumption would have no significant impact on the estimated drinking levels. However, in the scenarios of 50% and 75% screening coverage, the more conservative assumption would result in APC at the end of the ten-year simulation period to be, respectively, 5.2% (1.4 to 15.2%) and 6.8% (2.1 to 9.8%) higher.

## Discussion

### Principal findings

The impact of SBIRT delivered in German PHC settings on population level alcohol exposure was quantified using a newly developed simulation model. We built a time series of alcohol

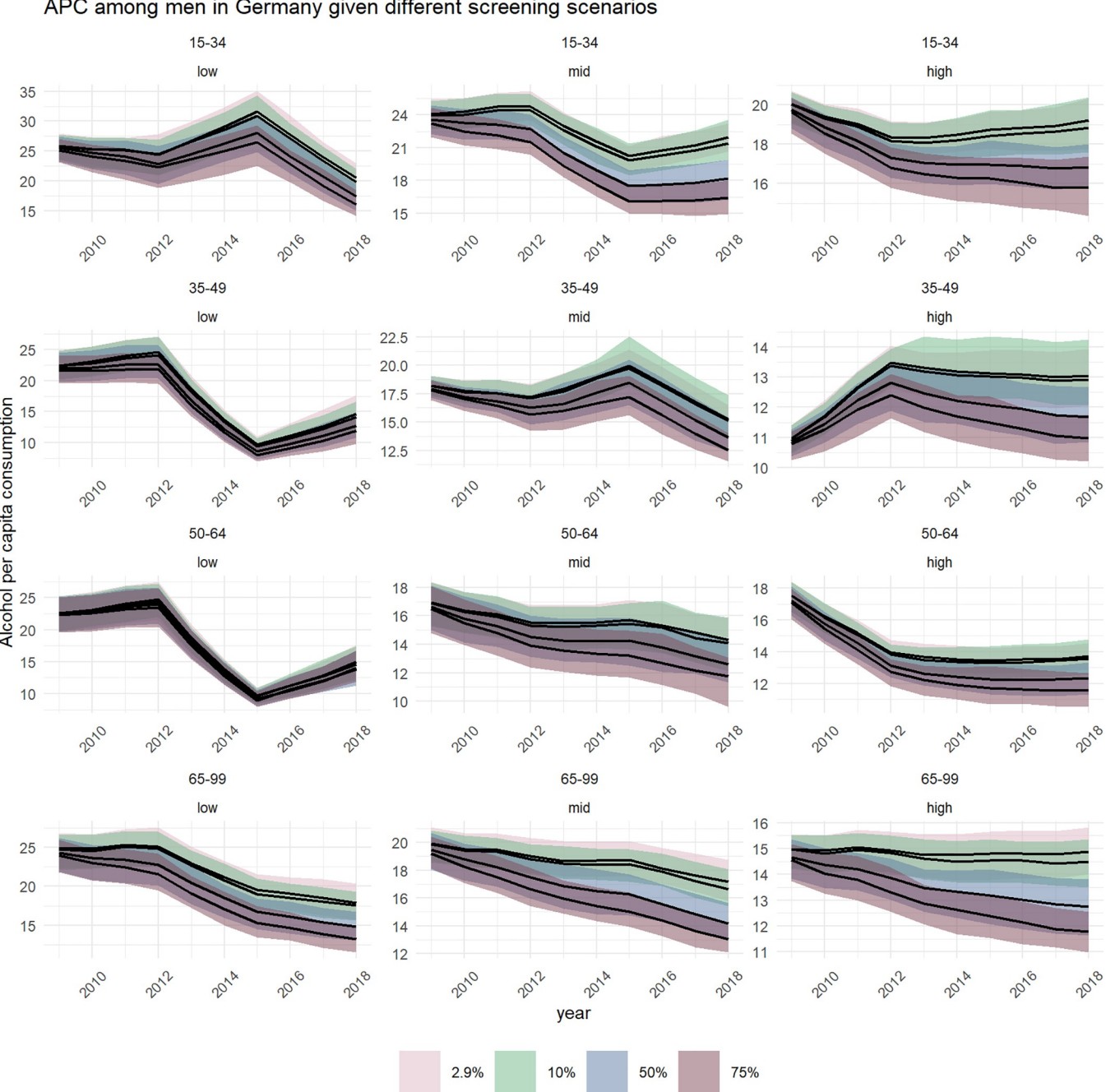

**Fig 5. Trends of alcohol per capita consumption (in litres pure alcohol) among German men between 2009 and 2018, stratified by age (horizontal) and education status (vertical), by annual screening coverage (coloured shade).** In each facet, the upper line represents the baseline scenario with a screening rate of 2.9%. The increasing coverage rate is represented by all lower lines with the second highest representing 10%, the third highest representing 50%, and the lowest representing 75% screening coverage rate.

exposure data by combining sales and survey data for the years 2009 to 2018 and assessed how alternative screening activities in PHC would have altered the observed trajectories. Our findings suggest that screening up to one tenth of patients per year would not have significantly changed how alcohol consumption has developed in Germany in this time period. Only if one fourth or more PHC patients had been screened for alcohol use once a year, a significant

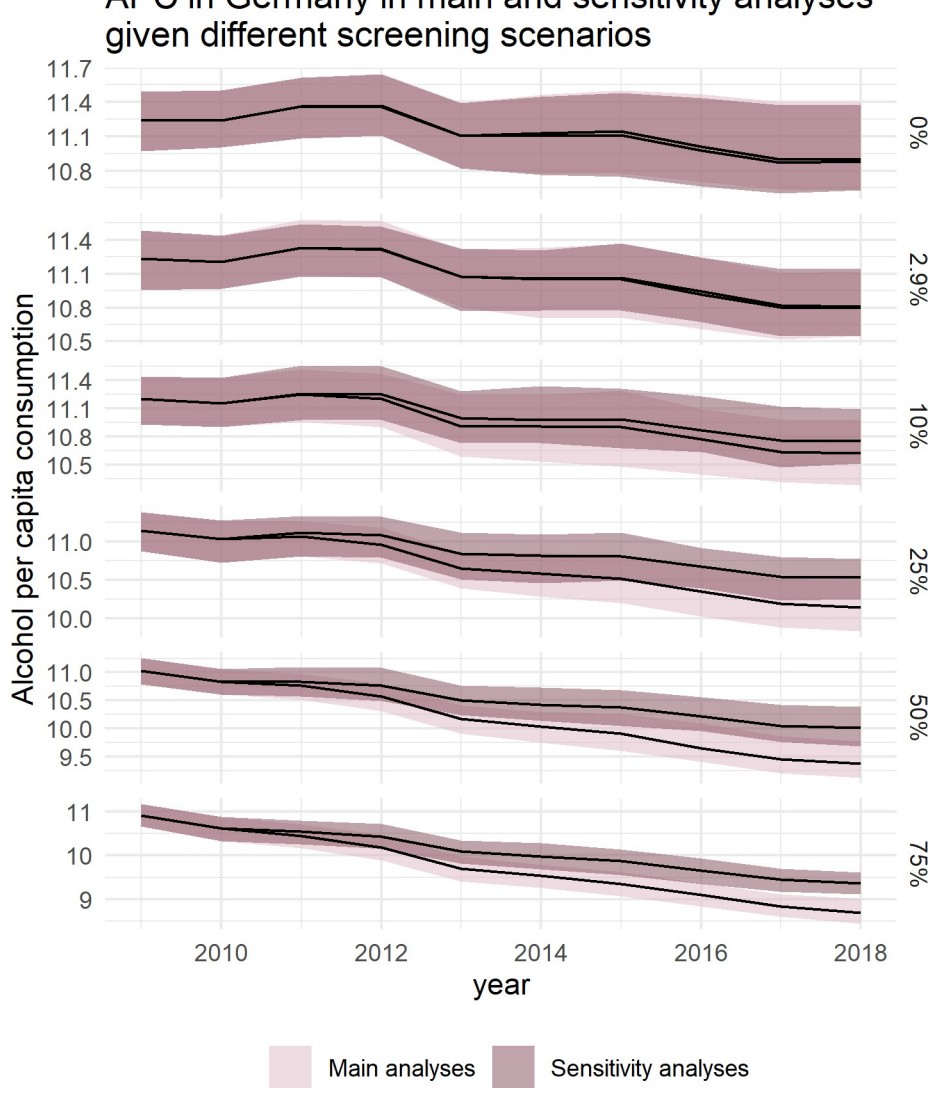

**Fig 6.** Trends of alcohol per capita consumption (in litres pure alcohol) in the German adult population between 2009 and 2018, by annual screening coverage (indicated at the right).

reduction over the ten-year period would have been observed in select groups. If every second PHC patient had been screened for alcohol use annually, per capita consumption would have been 11% lower in 2018.

## Limitations

Before further discussing the findings of this study, we need to highlight several limitations. First and foremost, as with any simulation study, we rely on assumptions that may not hold true. We have attempted to be transparent with all assumptions, reporting all parameters used in the simulation, and performing rigorous sensitivity analyses to test one key assumption. While simulation studies can provide important insights for policy planning, the results should always be treated with caution. Importantly, the larger the deviations from the as-is-scenario the larger the uncertainties. Second, we tried to rely mostly on local data, except for effect sizes

and attenuation parameters. Following this approach and given a lack of empirical data for Germany, we assumed the same probability for screening and BI for all drinkers, regardless of their socioeconomic position. However, BI reception was more often reported by socioeconomically disadvantaged drinkers in England [33], and if similar patterns were present in Germany, this would change the simulation findings accordingly. Third, we assume that effect sizes for BI and RT are similar, however, this might not be the case. People with very high drinking levels or with severe alcohol problems may not benefit from BI [34], however, recognition of their problems by PHC professionals through screening may result in the initiation of pharmacological and withdrawal treatment that may have greater effectiveness (for effects, see [35–37]). However, this implication is not considered in our simulation, as we followed a conservative approach in assuming that BI and RT effect sizes are similar.

## Application of the simulation model

We present an open-access simulation model that serves to estimate changes in alcohol consumption levels and patterns based on the implementation of alcohol interventions. Unlike most other simulation models, the entire source code and all input data are attached to this submission, enabling other researchers to adapt our work to other settings. Thus, this work prepares the ground for a number of applications, e.g., for estimating the potential of scaling up PHC-based alcohol interventions in other jurisdictions, or for assessing the impact of withdrawal interventions in inpatient settings (see e.g., [36]). If combined with health outcomes, e.g. by using the open-access programme InterMAHP [38], our simulation model can be readily adapted to perform health economic studies, such as cost-effectiveness analyses.

## Comparison with other simulation studies

Several other simulations have quantified the effects of scaling up SBIRT (for an overview, see [16]), however, we are only aware of one application for Germany. In a 2015 report, the Organization for Economic Co-operation and Development, the effects of BI were estimated for a screening coverage of 40% and a 30% intervention probability for positive screened patients, with the effects waning within 12 months after receiving the intervention [17]. In their simulation, which was performed for a 40-year time period, the prevalence of hazardous/harmful drinking could be reduced by 5%, while our results suggest a reduction in per capita consumption by 11% in the most comparable scenario of a 50% screening rate over a period of 10 years. As in our study, no measurable effects on HED for the total population were reported in the 2015 report.

Most existing simulation studies have aimed to assess the cost-effectiveness of implementing SBIRT in PHC settings (for an overview, see [16]). Our proposed simulation methodology can be extended as well to address health economic issues, including cost-effectiveness analyses. In contrast to most previous simulation studies, we publish the entire code and all input data alongside our results, encouraging other researchers to further develop and apply the proposed simulation model. Given the projected underachievement of achieving global goals in reducing alcohol consumption [14,39], in particular in Western European countries [40,41], applications of the simulation model may help to inform policy makers about the efforts required to achieve these targets. There are examples where it is possible to achieve high coverage of the measurement of alcohol consumption amongst PHC patients. In the UK, for example, an assessment of 1.8 million patient records in 2018 found that 48.8% of adult patients had a measure of alcohol consumption recorded during the previous five years [42]. Further, in integrated health-care systems where alcohol measurement is mandated and built into the

electronic medical record system, as it is in the US Veterans Health Administration system, coverage can be as high as 93% [43].

Why did we find little support for the impact of BI for reducing HED prevalence? According to the Cochrane review which provided the effect sizes for our simulation, only three out of ten trials found BI to reduce HED risk [24]. In fact, the only German trial on that matter also found no BI effects on HED status [44]. A number of literature reviews on that matter suggest that BI in PHC settings are most impactful among drinkers who are not dependent and not seeking treatment [45]. In this population, HED is not commonly reported [46], thus BI effects may be restricted to reduce the frequency of occasions with low to moderate amounts of alcohol intake.

## Implications for alcohol policy in Germany

The simulation results suggest that the current coverage of alcohol screening hardly matters for population alcohol exposure in Germany. While alcohol consumption is slowly declining, it remains among the highest in Europe [47] and globally [14]. We show that the large-scale delivery of SBIRT in German PHC settings could be a viable measure to accelerate the ongoing trend. However, in order to achieve this in Germany, a present lack of knowledge and awareness among PHC providers would need to be addressed. In a comparative survey, nearly half of German general practitioners did not consider alcohol as an important risk factor for hypertension, in contrast to a share of 15% among respondents from France, Italy, Spain, and the UK [48] (for evidence on alcohol use and hypertension, see [49]). Further, only every second general practitioner reported to be aware of the relevant alcohol management guideline in a recent survey [50]. Given the lack of improvement of alcohol management in PHC settings [51], alternative settings to implement SBIRT are considered. In a recent randomized-controlled trial, SBIRT delivery was tested in a municipal registry office in Germany responsible for registration, passport and vehicle admission issues, with no measurable effects on drinking behaviour [52].

While further efforts are needed to increase SBIRT delivery in German PHC settings in the long run, e.g., by financial reimbursement of alcohol management activities [53], alternatives may be required to reduce alcohol consumption and attributable burden in the short-term. In the light of very low alcohol taxation rates, including no taxes for wine [54], there is a large untapped potential in increasing retail prices for alcoholic beverages, which was demonstrated in two recent modeling studies [55,56]. Evidence from Lithuania and Great Britain, for example, demonstrates the impact that policies targeting alcohol prices can have in reducing consumption and harm [57–60].

## Conclusions

In Germany, alcohol consumption could have been 12% lower than it was in 2018 if every second PHC patient had been screened for alcohol since 2009. A large-scale implementation of SBIRT in Germany could only be achieved in the more distant future, thus, other alcohol policy options should be considered as well to achieve short-term reductions in alcohol consumption.

## Supporting information

**S1 Table. Alcohol consumption in the German adult population in 2009, by sex, age group and educational level.**
(DOCX)

**S1 File. R code and input data.**
(ZIP)

## Author Contributions

**Conceptualization:** Jakob Manthey.

**Data curation:** Jakob Manthey, Adriana Solovei, Sinclair Carr.

**Formal analysis:** Jakob Manthey, Sinclair Carr.

**Funding acquisition:** Peter Anderson, Jürgen Rehm.

**Investigation:** Jakob Manthey, Jürgen Rehm.

**Methodology:** Jakob Manthey, Peter Anderson, Jürgen Rehm.

**Project administration:** Jakob Manthey, Jürgen Rehm.

**Resources:** Jakob Manthey, Adriana Solovei, Peter Anderson, Sinclair Carr, Jürgen Rehm.

**Software:** Jakob Manthey.

**Supervision:** Jakob Manthey, Peter Anderson, Jürgen Rehm.

**Validation:** Jakob Manthey, Peter Anderson.

**Visualization:** Jakob Manthey, Adriana Solovei.

**Writing – original draft:** Jakob Manthey.

**Writing – review & editing:** Jakob Manthey, Adriana Solovei, Peter Anderson, Sinclair Carr, Jürgen Rehm.

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
