## [Decision Letter · Decision Letter 0]

9 Jun 2021

PONE-D-21-11622

Can alcohol consumption in Germany be reduced by alcohol screening, brief intervention and referral to treatment in primary health care? Results of a simulation study

PLOS ONE

Dear Dr. Manthey,

Thank you for submitting your manuscript to PLOS ONE. After careful consideration, we feel that it has merit but does not fully meet PLOS ONE’s publication criteria as it currently stands. Therefore, we invite you to submit a revised version of the manuscript that addresses the points raised during the review process.

The revised version should address all comments.

We look forward to receiving your revised manuscript.

Kind regards,

Petri Böckerman

Academic Editor

PLOS ONE

Journal Requirements:

Additional Editor Comments (if provided):

The revised version should address all comments.

Reviewers' comments:

Reviewer's Responses to Questions

**Comments to the Author**

1. Is the manuscript technically sound, and do the data support the conclusions?

Reviewer #1: Yes

Reviewer #2: Yes

2. Has the statistical analysis been performed appropriately and rigorously? 

Reviewer #1: Yes

Reviewer #2: Yes

3. Have the authors made all data underlying the findings in their manuscript fully available?

Reviewer #1: Yes

Reviewer #2: Yes

4. Is the manuscript presented in an intelligible fashion and written in standard English?

Reviewer #1: Yes

Reviewer #2: Yes

5. Review Comments to the Author

Reviewer #1: Dr. Manthey and collaborators investigated whether the implementation of an alcohol screening, during the primary health care visit, and the brief intervention and/or referral treatment would reduce the amount of drinking levels and heavy episodic drinking in Germany population. Authors used a very robust model to simulate different scenarios (based in the percentage of patients screened). They controlled the data for sex, age, and educational levels, and also counted for the under-reported drinking levels, since the data input was acquired from different surveys. Their simulation showed that if 50% of the primary health care patients would be screened for alcohol use, the consumption of this drug could be reduced in 12%. Authors, also highlights the importance of making their model open access as a tool for other studies. In general, the study is timely, the bioinformatic tools applied are sound, the authors provide clear models and hypothesis, and the findings are clearly embedded in a theoretical framework in the discussion section.

The figures resolution needs to be improved. I could not read the diagram in figure 1. Authors could add to the discussion their perspective about why the screening is not efficient to reduce the heavy episodic drinking

Reviewer #2: The study is well-designed and the methodology is frankly conveyed. The authors' expertise is fully evident. The paper has evident policy implications, suggesting that other alcohol policies besides SBIRT likely are needed in order to achieve short-term reduction of alcohol use and related harms in Germany.

6. PLOS authors have the option to publish the peer review history of their article (what does this mean?). If published, this will include your full peer review and any attached files.

Reviewer #1: No

Reviewer #2: No

---

## [Author Response · Author response to Decision Letter 0]

17 Jun 2021

Reviewer #1: 

Dr. Manthey and collaborators investigated whether the implementation of an alcohol screening, during the primary health care visit, and the brief intervention and/or referral treatment would reduce the amount of drinking levels and heavy episodic drinking in Germany population. Authors used a very robust model to simulate different scenarios (based in the percentage of patients screened). They controlled the data for sex, age, and educational levels, and also counted for the under-reported drinking levels, since the data input was acquired from different surveys. Their simulation showed that if 50% of the primary health care patients would be screened for alcohol use, the consumption of this drug could be reduced in 12%. Authors, also highlights the importance of making their model open access as a tool for other studies. In general, the study is timely, the bioinformatic tools applied are sound, the authors provide clear models and hypothesis, and the findings are clearly embedded in a theoretical framework in the discussion section.

The figures resolution needs to be improved. I could not read the diagram in figure 1. Authors could add to the discussion their perspective about why the screening is not efficient to reduce the heavy episodic drinking

Response:

Thank you for your positive assessment of our work. We have improved the resolution of Figure 1 to 300 DPI. Further, we have added a paragraph explaining possible reasons why we found little support for reducing heavy episodic drinking (HED):

“Why did we find little support for the impact of BI for reducing HED prevalence? According to the Cochrane review which provided the effect sizes for our simulation, only three out of ten trials found BI to reduce HED risk [24]. In fact, the only German trial on that matter also found no BI effects on HED status [44]. A number of literature reviews on that matter suggest that BI in PHC settings are most impactful among drinkers who are not dependent and not seeking treatment [45]. In this population, HED is not commonly reported [46], thus BI effects may be restricted to reduce the frequency of occasions with low to moderate amounts of alcohol intake.”

Reviewer #2: 

The study is well-designed and the methodology is frankly conveyed. The authors' expertise is fully evident. The paper has evident policy implications, suggesting that other alcohol policies besides SBIRT likely are needed in order to achieve short-term reduction of alcohol use and related harms in Germany.

Response: Many thanks for your positive assessment of our work.

---

## [Decision Letter · Decision Letter 1]

26 Jul 2021

Can alcohol consumption in Germany be reduced by alcohol screening, brief intervention and referral to treatment in primary health care? Results of a simulation study

PONE-D-21-11622R1

Dear Dr. Manthey,

We’re pleased to inform you that your manuscript has been judged scientifically suitable for publication and will be formally accepted for publication once it meets all outstanding technical requirements.

Kind regards,

Petri Böckerman

Academic Editor

PLOS ONE

Additional Editor Comments (optional):

Reviewers' comments:

Reviewer's Responses to Questions

**Comments to the Author**

1. If the authors have adequately addressed your comments raised in a previous round of review and you feel that this manuscript is now acceptable for publication, you may indicate that here to bypass the “Comments to the Author” section, enter your conflict of interest statement in the “Confidential to Editor” section, and submit your "Accept" recommendation.

Reviewer #1: All comments have been addressed

Reviewer #2: All comments have been addressed

2. Is the manuscript technically sound, and do the data support the conclusions?

Reviewer #1: Yes

Reviewer #2: (No Response)

3. Has the statistical analysis been performed appropriately and rigorously? 

Reviewer #1: Yes

Reviewer #2: (No Response)

4. Have the authors made all data underlying the findings in their manuscript fully available?

Reviewer #1: Yes

Reviewer #2: (No Response)

5. Is the manuscript presented in an intelligible fashion and written in standard English?

Reviewer #1: Yes

Reviewer #2: (No Response)

6. Review Comments to the Author

Reviewer #1: The authors addressed all my questions and added all additional information requested. Overall, I think this is an excellent study, that is well written, and well presented. The conclusions are solid, and do not reach beyond the data.

Reviewer #2: It is a very solid paper. The study is well-designed and the methodology is frankly conveyed. The authors'

expertise is fully evident. The paper has evident policy implications, suggesting that

other alcohol policies besides SBIRT likely are needed in order to achieve short-term

reduction of alcohol use and related harms in Germany.

7. PLOS authors have the option to publish the peer review history of their article (what does this mean?). If published, this will include your full peer review and any attached files.

Reviewer #1: No

Reviewer #2: No

---

## [Editor Report · Acceptance letter]

28 Jul 2021

PONE-D-21-11622R1 

Can alcohol consumption in Germany be reduced by alcohol screening, brief intervention and referral to treatment in primary health care? Results of a simulation study 

Dear Dr. Manthey:

I'm pleased to inform you that your manuscript has been deemed suitable for publication in PLOS ONE. Congratulations! Your manuscript is now with our production department. 

Kind regards, 

on behalf of

Professor Petri Böckerman 

Academic Editor

PLOS ONE